# Subsets of Cytokines and Chemokines from DENV-4-Infected Patients Could Regulate the Endothelial Integrity of Cultured Microvascular Endothelial Cells

**DOI:** 10.3390/pathogens11050509

**Published:** 2022-04-26

**Authors:** Marcio da Costa Cipitelli, Iury Amancio Paiva, Jéssica Badolato-Corrêa, Cíntia Ferreira Marinho, Victor Edgar Fiestas Solórzano, Nieli Rodrigues da Costa Faria, Elzinandes Leal de Azeredo, Luiz José de Souza, Rivaldo Venâncio da Cunha, Luzia Maria de-Oliveira-Pinto

**Affiliations:** 1Viral Immunology Laboratory, Instituto Oswaldo Cruz—Fiocruz, Rio de Janeiro 21040-360, Brazil; marcio_cipitelli@msn.com (M.d.C.C.); iury.iap@gmail.com (I.A.P.); jessicabadolato04@gmail.com (J.B.-C.); cintiafm11@gmail.com (C.F.M.); vicfiso@gmail.com (V.E.F.S.); elzinandes@ioc.fiocruz.br (E.L.d.A.); 2Flavivirus Laboratory, Instituto Oswaldo Cruz—Fiocruz, Rio de Janeiro 21040-360, Brazil; nielircf@gmail.com; 3Centro de Referência de Doenças Imuno-Infecciosas, Campos dos Goytacazes 28025-496, Brazil; luizjosedes@gmail.com; 4Departamento de Clínica Médica, Faculdade de Medicina de Campos, Campos dos Goytacazes 28035-581, Brazil; 5Faculdade de Medicina, Universidade Federal de Mato Grosso do Sul, Campo Grande 79070-900, Brazil; rivaldo.cunha@fiocruz.br

**Keywords:** dengue, cytokines, chemokines, endothelial cells, permeability

## Abstract

**Introduction:** It is a consensus that inflammatory mediators produced by immune cells contribute to changes in endothelial permeability in dengue. We propose to relate inflammatory mediators seen in dengue patients with the in vitro alteration of endothelial cells (ECs) cultured with serum from these patients. **Methods:** Patients with mild (DF) to moderate and severe dengue (DFWS/Sev) were selected. ELISA quantified inflammatory mediators. Expression of adhesion molecules and CD147 were evaluated in the ECs cultured with the patient’s serum by flow cytometry. We assessed endothelial permeability by measuring transendothelial electrical resistance in cocultures of ECs with patient serum. **Results:** Dengue infection led to an increase in inflammatory mediators—the IL-10 distinguished DF from DFWS/Sev. There were no changes in CD31, CD54, and CD106 but decreased CD147 expression in ECs. DFWS/Sev sera induced a greater difference in endothelial permeability than DF sera. Correlation statistical test indicated that low IL-10 and IFN-γ and high CCL5 maintain the integrity of ECs in DF patients. In contrast, increased TNF, IFN-γ, CXCL8, and CCL2 maintain EC integrity in DFWS/Sev patients. **Conclusions:** Our preliminary data suggest that a subset of inflammatory mediators may be related to the maintenance or loss of endothelial integrity, reflecting the clinical prognosis.

## 1. Introduction

The vascular endothelium is involved in different processes, including regulating the selective exchange of macromolecules and cells between blood and tissue fluids, coagulation, induction of inflammatory responses, and angiogenesis. The endothelium also responds to the microenvironment by synthesizing and metabolizing products that act autocrine and paracrine to maintain homeostasis [1,2]. For this, the permeability of vessels to elements present in the circulation is highly coordinated and controlled by transcellular and paracellular transport pathways, which integrate the rearrangement of intercellular junctions and the cell cytoskeleton [3]. There is a dynamic control, highly sensitive to changes in the vascular microenvironment, which regulates the speed of macromolecule transport, surveillance of immune cells, and deposition of extracellular matrix (ECM) proteins close to the vessel wall in the tissue repair process [4]. The endothelial barrier, in general, maintains vascular permeability through the glycocalyx, integrins expressed on the cell surface such as αvβ3, αvβ1, adhesion molecules, such as CD31/PECAM, CD54/ICAM, and CD106/VCAM, and adhesion molecules, such as VE-cadherin [5]. In pathological conditions such as infections, an increase in the concentrations of cytokines, chemokines, nitric oxide, and growth factors, in addition to the activation of the coagulation cascade, can cause dysfunction of the endothelial barrier [6,7,8,9].

Severe and lethal cases of dengue patients are usually caused by an increase in vascular permeability that leads to the accumulation of fluid in the cavities, reduction of blood pressure, and impairment of organ perfusion. [10]. Analysis of different necropsy tissues from dengue patients identified the viral antigen. However, the authors did not observe infected endothelial cells (ECs) or greater morphological damage to the endothelium [11,12]. In vivo studies suggest that ECs are poorly infected; therefore, they are not important targets for DENV replication [13,14,15]. Singh et al. [16] explored the relationship between the infectivity phenotype and the ability to induce the role of transendothelial leakage among DENV serotypes. The authors compared two closely related DENV serotype-2 strains of the Cosmopolitan genotype. They found that the less infectious strain induces more leakage of trans-endothelial cells into the ECs monolayer after infection and secretes more non-structural protein (NS1) into the culture supernatant. Their data support the vascular leakage-inducing potential of DENV strains linked to sNS1 levels [16]. Later, many authors demonstrated different NS1-related mechanisms. NS1 was seen to activate toll-like receptor-4 signaling in primary human myeloid cells, leading to pro-inflammatory cytokine secretion and vascular leakage [17]. Furthermore, NS1 can disrupt the glycocalyx layer in human pulmonary microvascular ECs, inducing sialic acid degradation and the release of heparan sulfate from proteoglycans [18]. Recently, macrophage migration inhibitory factor (MIF) has been directly involved in NS1-induced glycocalyx degradation and hyperpermeability via activation of heparinase-1 and metalloproteinase (MMP) 9 [19].

In addition to NS1, in vitro studies have shown that DENV infects human umbilical ECs through cellular receptors containing heparan sulfate. Infected ECs release several soluble mediators, including interleukin (IL)-6, CXCL8/IL-8, CXCL9/MIG, CXCL10/IP-10, and CXCL11/I-TAC, all of which are involved in both activating the antiviral immune response and increasing the endothelial permeability [20]. Of note is TNF-α, which has been closely associated with vascular leakage [21,22]. Inyoo et al. suggested a synergistic role of TNF-α in increasing endothelial permeability, leading to changes in adhesion junctions and tight junctions in DENV-infected human ECs [23]. Taken together, the effect of TNF-α and other molecules on the loss of vascular integrity and function may explain extensive damage to the endothelium-independent of a major in vivo infection of the endothelium. The vasoactive endothelial factors released from the infected cells would contribute to the occurrence of this phenomenon. In addition to ECs, monocytes/macrophages, and dendritic cells (DCs) are targets for viral replication. All these target cells are producers of IL-1, IL-6, MIF, TNF-α, and MMPs [24,25]. EC barrier alteration study using in vitro models described soluble factors released from infected macrophages alter ECs permeability in the absence of virus-induced cytopathic effect [26].

In the present study, we quantified a set of inflammatory mediators in the blood of dengue patients with different clinical outcomes. Next, we asked whether these mediators, when placed in culture with a monolayer of endothelial cells, HMVEC-d, could regulate the expression of adhesion molecules CD31, CD54, and CD106 and the molecule CD147, and change the permeability of these ECs. We hope to extrapolate the data obtained to propose a set of inflammatory mediators that would be potential biomarkers of endothelial permeability and vascular leakage changes in patients with dengue.

## 2. Results

### 2.1. Characteristics of Cohort Dengue Patients and Healthy Donors

Dengue patients were classified according to the World Health Organization’s TDR (Special Programme for Research and Training in Tropical Diseases) [27]. Of these, 49 are dengue fever without warning signs (DF), 16 are DF with warning signs (DFWS), and 5 as severe dengue (Sev). A total of 10 healthy donors (HD) showed no signs of infection, fever or malaise, or any chronic illness.

Regarding the days of illness counted from the first signs/symptoms until the date of sample collection, the DF had a shorter time of disease evolution than the DFWS/Sev.

All dengue patients had headaches, fever, retro-orbital pain, myalgia, arthralgia, anorexia, and nausea. DFWS/Sev individuals present with DF-like signs/symptoms, including persistent abdominal pain, and vomiting. A total of 35% of DFWS/Sev patients had some symptoms related to vascular leakage, including hepatosplenomegaly, fluid accumulation, and organ dysfunction. Still, 32% of them had mucosal and gingival bleeding, hematuria, petechiae, or melena.

RT-PCR identified the viral serotype in 48.8% of DF and 21.1% of DFWS/Sev patients. DENV-4 is the most prevalent among dengue groups. Diagnostic methods confirmed 41.7% of DF and 95.2% pf DFWS/Sev patients using anti-DENV IgM. The presence of the NS1 antigen in the blood was seen in 64.4% and 50% of DF and DFWS/Sev, respectively. Considering the importance of measuring NS1, our group established the protocol to quantify NS1 in patient samples [28]. We measured NS1 (ng/mL) in our cohort using this protocol. Still, we found no differences between the DF and DFWS/Sev groups.

Thrombocytopenia is frequent in dengue, so we observed a lower platelet count in dengue patients than in healthy donors. We did not observe any statistical difference between the hematocrit percentages of patients and healthy donors, which can be explained by the fact that patients are generally rapidly submitted to intravenous hydration, reducing hemoconcentration. We did not observe any statistical difference in transaminases AST and ALT between dengue patients and healthy donors.

The total leukocyte count was lower in DF and DFWS/Sev when compared to healthy donors, but we did not observe a statistical difference between dengue groups. As for lymphocyte counts, DFWS/Sev had lower counts than healthy donors. On the other hand, the monocyte count was higher in DF than DFWS/Sev. We did not observe any statistical difference between patients and donors. All patients and donors’ characteristics are depicted in Table 1.

### 2.2. Increase in Inflammatory Mediators in Dengue Patients Compared to Healthy Donors

An extensive literature points to the effects of inflammatory mediators in altering endothelial permeability. Thus, we initially characterized the profile of inflammatory mediators in our cohort of dengue patients. With this data set in hand, we will be able to assess the relationship between these mediators and changes in HMVEC-d.

The data demonstrate that IFN-γ and CCL5/RANTES levels did not differentiate dengue groups and healthy donors. However, regardless of the clinical outcome, both DF as DFWS/Sev patients had elevated TNF-α, CXCL8/IL-8, CX3CL1/Fractalkine, CXCL10/IP-10, and CCL2/MCP-1 compared to healthy donors. IL-10 levels were significantly higher in dengue groups compared to healthy donors and, still, IL-10 differentiate DF from DFWS/Sev patients (Figure 1).

### 2.3. Decreased Expression of CD147/EMMPRIN in HMVEC-d Cultured in the Presence of Serum from DENV-Infected Patients Who Also Have Reduced Levels of MMP-9

Our next question was to evaluate whether the addition of serum from dengue patients on a monolayer of endothelial cells would be able to change the expression of molecules involved in the functioning of these cells. For this, HMVEC-d were seeded in transwell inserts and cultured in the presence of randomly chosen sera of 11 DF and 10 DFWS/Sev (8 DFWS and 2 Sev) patients. HMVEC-d in culture medium was used as a control. We evaluated the expression of CD31/PECAM-1, CD54/ICAM-1, and CD106/VCAM-1 adhesion molecules, all involved in the leukocyte transmigration process [29], and CD147/EMMPRIN, which leads to the production of matrix metalloproteinases [30].

We demonstrated the analysis strategy in Figure 2A. A population of viable cells was selected based on their morphological characteristics (FSC size—forward scatter and SSC granularity—side scatter). As can be seen, HMVEC-d constitutively expresses high levels of CD147/EMMPRIN and CD31/PECAM-1, but the little expression of CD54/ICAM-1 and CD106/VCAM-1 (Figure 2B).

In cultures of HMVEC-d in which patient sera were added, we did not observe changes in CD31/PECAM-1 (Figure 2C), CD106/VCAM-1 (Figure 2D), and CD54/ICAM-1 (Figure 2E) in endothelial cells. However, DF and DFWS/Sev sera decreased CD147/EMMPRIN expression on HMVEC-d (Figure 2F). We provide Appendix A with the raw data of the points present in Figure 2C–F.

The patients’ cytokine and chemokine measurements were correlated with CD147 expression on HMVEC-d. From these analyses, CX3CL1 from patients with DF (r = 0.833, *p* < 0.02) and TNF-α from DFWS/Sev (r = 0.663, *p* < 0.05) patients was directly associated with CD147 (Appendix A). It was impossible to apply correlation tests for cytokines IL-10, CX3CL1 and CXCL10 from DFWS/Sev patients because the number of evaluations was small.

Interestingly, among several MMPs related to CD147 activation, we measured MMP-9 in the serum of dengue patients. To our surprise, the levels of MMP-9 are low in dengue patients compared to healthy donors (Figure 2G).

### 2.4. Cytokines and Chemokines Present in the Serum of Dengue Patients Can Lead to Changes in the Permeability of Endothelial Cells That Are More Frequent in DFWS/Sev Patients Than in DF

Finally, we analyzed whether serum samples from dengue patients would alter the permeability of the endothelial cell monolayer. HMVEC-d were seeded on transwell inserts and cultured with serum from subjects. Serum from healthy donors were used as controls. Eight DF and 8 DFWS/Sev (6 DFWS and 2 Sev) were randomly chosen. We measured transendothelial electrical resistance (TEER) at 5, 15, and 30 min and 1, 2, 3, and 4 h. In some individuals, we kept the TEER reading up to 24 h. However, the cells did not look healthy at this point. Thus, we excluded the 24 h point from the analysis to avoid misinterpretations. All TEER checkpoints were compared to time 0, defined as the time before the addition of serum. At time 0, we assumed the relative TEER values to be 1.0, and all TEER values were normalized as described in Materials and Methods. According to our findings, in the five healthy donors, no significant change was detected in the relative TEER values after the addition of serum when compared to time 0 (Figure 3A). We also did not find significant differences between the relative TEER values before and after the addition of serum to DF (Figure 3B) or DFWS/Sev patients (Figure 3C).

An additional analysis was realized comparing healthy donors, DF and DFWS/Sev patients in each timepoint of relative TEER values but no significant change was detected (Figure 3D).

To better understand our data, we established that relative TEER values less than or equal to 0.4 (discriminated in gray) constituted a “critical zone”, as all our healthy donors had relative TEER values above 0.4 (Figure 3A–D). We saw that at least two of the eight DF patients, or 25% of them, had a relative TEER value ≤0.4 at some time in the kinetics. Regarding the DFWS/Sev, six of the eight or 75% had a relative TEER value ≤0.4 at some time kinetics. This difference was statistically significant according to Chi-square analysis (*p* < 0.05).

Our next question was: which cytokines and chemokines present in the patients’ sera would be associated with changes in the permeability of endothelial cells? We applied statistical correlation analysis to assess possible interactions of relative TEER values with serum amounts of cytokines/chemokines.

Interestingly, low levels of IFN-γ and IL-10 in DF patients seem to care for the integrity of the endothelial cell monolayer since we detected a strong inverse correlation between both cytokine levels and the relative TEER values. On the other hand, high levels of CCL5 appear to protect the endothelial cell monolayer, as CCL5 is directly correlated with relative TEER (Figure 3E). In DFWS/Sev patients, increased levels of IFN-γ and TNF-α appear to maintain the integrity of the endothelial cell monolayer, which could be further synergized by high levels of CXCL8/IL-8 and CCL2/MCP-1 (Figure 3F). These data encourage us to study the potential in vivo effect of cytokines and chemokines as biomarkers of alteration of endothelial cell monolayers in dengue patients.

### 2.5. Are There Differences between Patients with a Relative TEER Value >0.4 Versus ≤0.4?

Surprisingly, one DFWS patient and one Sev patient did not show a decrease in the relative value of TEER (TEER ≤ 0.4). On the other hand, two DF did (TEER ≤ 0.4). Thus, we performed further statistical analyzes to understand whether there are differences considering the “critical zone”, that is, patients with TEER > 0.4 versus those with TEER ≤ 0.4 (Table 2). All possible parameters were analyzed based on this classification. We also measured NS1 (ng/mL) in dengue patients reclassified based on TEER critical zone, and we also found no difference between groups. Interestingly, hematocrit percentage and monocyte count were higher in patients with TEER > 0.4 than those with TEER ≤ 0.4. We observed a trend (assuming *p*-value < 0.1) of higher levels of CXCL8/IL-8 and CCL5/RANTES in patients with TEER > 0.4 than in patients with TEER ≤ 0.4.

## 3. Discussion

When we talk about dengue and vascular endothelium, two sides of the coin come to mind. If, on the one hand, the alteration of the endothelium is a crucial step for the transendothelial migration of leukocytes, a fundamental process for the activation of immunity, inflammation, and effective immune control of infectious agents such as DENV; on the other hand, the loss of vascular integrity in DENV infection is characteristic of an immunopathological response, especially in severe forms of dengue, as it contributes to bleeding and damage to organs.

Our hypothesis is that a set of inflammatory mediators present in the blood of dengue patients with different clinical outcomes can differentially regulate the expression of adhesion molecules CD31, CD54, and CD106 and the CD147 molecule, in addition to altering the permeability of ECs in in vitro assays. From this information, we could present and contribute to a panel of biomarkers of plasma leakage and poor clinical prognosis.

A recent review by Malavige et al. [31] discussed different and complex mechanisms of the balance between an immunoprotective versus immunopathogenic response in dengue. Among them, a vast panel of increased cytokines and chemokines in severe dengue, such as IFN-γ, GM-CSF, IL-10, CCL4/MIP-1β, IL-1β, CXCL8/IL-8 TNF-α, CXCL10/IP-10, CCL2/MCP-1, and IL-18. Others, such as TNF-α and IL-1β, directly cause vascular leakage, IL-18, CXCL10/IP-10, CXCL8/IL-8, and CCL4/MIP-1β, which are potent inflammatory cytokines produced by numerous immune cells.

In our study, we initially characterized the clinical and laboratory parameters of the DF and DFWS/Sev groups and then measured the serum levels of cytokines, chemokines, and MMP-9 in the groups. We designed this study to evaluate the effect of adding serum from infected patients to human ECs cultures. We then evaluated the expression of adhesion molecules and the CD147 molecule and measured the transendothelial electrical resistance (TEER) under the above conditions. In particular, we have a cohort of a few severe dengue cases, which was the epidemiological scenario at the time of the study. As far as we know, few studies have carried out experimental approaches like ours. Most published studies use DENV-infected ECs in vitro [13,32,33] or evaluate the addition of NS1 protein in ECs cultures. Another uses samples from patients with substantial plasma leakage or severe symptomatic dengue [34,35]. All of them have potentially contributed to the understanding of the mechanisms involved in the alteration of endothelial permeability in dengue. However, in cases of the less severe disease, we do not know whether the detected alterations in the inflammatory profile of these patients are sufficient to induce some alteration in the ECs in vitro.

Previous data from our group showed that, regardless of the infecting DENV serotype, viremia and NS1 levels are higher in fatal than in non-fatal cases. Thus, viremia and NS1 antigenemia appear to be potential biomarkers of dengue severity [36]. Herein, we do not know whether the in vivo amounts of NS1 measured in our patients would be sufficient to induce changes in endothelial permeability.

Many scientific groups have already demonstrated the presence of several autoantibodies directed against autoantigens expressed in platelets, endothelial cells, and clotting proteins in dengue [37,38,39,40]. Recently, the group of Vo et al. [41] showed an increase of 80 autoantibodies of the IgM class and six of the IgG class in the plasma of dengue patients compared to healthy donors. The authors used a protein matrix assay to analyze 128 putative autoantigens to screen for IgM and IgG reactivity in patients. But despite the enormous contribution of Vo’s data, they did not conduct experiments to assess the cross-reactivity of these autoantibodies to viral antigens. But Cheng et al. [42] showed that anti-DENV NS1 IgM and IgG antibodies cross-react with endothelial cells. Among the identified epitopes, the authors detected elevated levels of IgM and IgG against the NS1 311-330 epitope in patients with dengue hemorrhagic fever compared to controls. The NS1 311-330 epitope (P311-330) is a cross-reactive epitope against protein disulfide isomerase (PDI) expressed on endothelial cells. In our study, even without exploring the effects of anti-DENV IgM, we observed a difference in the frequency of positivity of anti-DENV IgM between the DF and DFWS/Sev groups (41% vs. 95%, respectively). Whether these anti-IgM DENV interfere in the pathogenesis of dengue are question to be explored.

Our study used a cohort of a few cases of severe dengue, most of them infected with the DENV-4 serotype. Dengue has been a public health problem in Brazil, since DENV-1 introduction in the ’80s. The scenario was aggravated by the introduction of DENV-2 and DENV-3 in the ’90s and 2000s, respectively. After that, those serotypes co-circulated causing epidemics in many regions over the years. However, DENV-4 was introduced in 2010 in the North region and rapidly spread throughout the country. Due to the populations’ susceptibility to this newly introduced serotype, explosive epidemics occurred and in 2012, DENV-4 was prevalent and detected in 63% of the cases reported in Brazil. Despite the highest notification in 2013 (1,452,289 cases), DENV-4 circulation was associated with mild cases, but severe and fatal cases due to this serotype were also reported [43]. In Rio de Janeiro, the virus was first isolated in 2011 and resulted in the emergence of this serotype in the state in 2012, characterized by isolation of this serotype in 48.7% of confirmed cases and, in 2013, DENV-4 was responsible for the largest number of cases in the state [28]. Despite being known as a mild serotype, the impact of the emergence of DENV-4 in an endemic region where the other three serotypes were circulating was unknown. The introduction of a new serotype of DENV is associated with the occurrence of major epidemics and increased proportion of severe cases [44], more frequent in individuals susceptible to the new serotype circulating.

A previous study showed that DENV-4, compared to other serotypes, induced the least changes in endothelial permeability using four different human endothelial microvascular cell lines (brain, dermal, pulmonary, and retinal). According to the authors, ECs were differentially activated soon after viral infection, showing a temporary change in permeability in the dermal origin or a long-term change in the pulmonary, retinal, and cerebral origin [45,46,47].

As mentioned, we used a cohort of patients clinically characterized as DF, DFWS, and few cases of severe dengue. Both groups could be well defined in terms of clinical signs/symptoms and agreed with expectations regarding the days after the onset of symptoms, which was higher in DFWS/Sev. The DFWS/Sev was also the one that presented greater vascular alterations and mild hemorrhagic conditions. The production profile of inflammatory mediators did not distinguish the two groups, except for IL-10, which was higher in DFWS/Sev than in DF. Despite numerous published works, the ultimate role of IL-10 in different diseases has not been fully determined, due to the extremely heterogeneous immunological contexts that regulate its functions and the controversial documented effects [46,47,48]. In COVID-19, authors demonstrated an increase in mRNA of ACE2, SARS-CoV2 receptor, in pulmonary cells, and in ECs in vitro after treatment with IL-10. Thus, IL-10 upregulation may be clinically relevant in acute respiratory distress syndrome and vasculitis associated with COVID-19 [49].

As dengue and COVID-19 share immunopathogenic mechanisms, a recent study evaluated the similarities and differences in cytokine and chemokine responses in patients with varying severity of COVID-19 and acute dengue. Of the cytokines evaluated, IL-10 was significantly higher in patients with severe COVID-19 pneumonia and in those who developed dengue hemorrhagic fever (DHF) compared with DF, indicating IL-10 as a biomarker of an altered antiviral response potentially contributing to disease severity [50]. In this way, we suggest that IL-10, even in mild to moderate cases of dengue, is a marker that should be investigated to help the clinical prognosis.

The endothelium is an important replicative niche for viruses and other pathogens [51]. Although ECs secrete inflammatory mediators [52], the endothelial barrier is also affected by these mediators that are secreted by immune cells, including monocytes, macrophages, DCs, and T lymphocytes [53,54]. A much earlier study than ours demonstrated that supernatants from DENV-2-infected monocyte-derived macrophages could increase permeability in EC monolayers after the peak of progeny virus release and independent of TNF-α release. Thus, in 2003, authors concluded that supernatants from DENV-2-infected monocyte-derived macrophages contain many factors that increase the permeability of ECs [55]. In contrast, addition of the supernatant to the EC monolayer HMVECs promoted change in the permeability of the monolayer by decreasing the measurement of the TEER, reversed after blocking of the TNF-α that recovery of monolayer integrity [56].

Besides TNF-α, CXCL8/IL-8 is also an important vasoactive mediator [56]. Such proinflammatory mediators promoted increased expression, activation, and deregulation of adhesion molecules (PECAM-1/CD31, ICAM-1/CD54, and VCAM-1/CD106) in ECs. Additionally, they may play a role in protein modification of the cytoskeleton or tight junctions [45]. In vitro assays using CXCL8/IL-8 and CCL2/MCP-1, in general present at high levels in DHF and dengue shock syndrome (DSS) patients, influenced the transendothelial permeability of EC lines by regulating their tight junctions and cytoskeleton. DENV-infected monocyte supernatants pretreatment with CXCL8/IL-8 and CCL2/MCP-1 neutralizing antibody resulted in a partial reversal of permeability change in ECs. It indicates the role of such chemokines in the process [13,33]. The platelet-activating factor (PAF), MIF, MMP-9, and CCL2/MCP-1 reduced PECAM-1 and vascular endothelial (VE)-cadherin expression on the endothelial membrane by disrupting the distribution of tight junction ZO-1 protein, and by redistributing F-actin fiber [33,34,35,56]. Altogether, a stronger inflammatory response may potentially be a molecular basis that accounts for increased endothelial permeability or vascular leakage in dengue.

Herein, HMVEC-d pretreated with patient sera apparently did not demonstrate any change in the CD31/PECAM-1, CD106/VCAM-1, and CD54/ICAM-1 expressions on endothelial cells. These results were unexpectable for us once high levels of soluble ICAM-1/CD54 and VCAM-1/CD106 were already demonstrated in dengue patients with vascular damage [56]. However, certainly other molecules expression could have been altered. In this context, besides glycocalyx, Kanlaya et al. performed functional alteration analyses in actin cytoskeletal assembly and endothelial integrity focusing on adherens junction (VE-cadherin) and, tight junction (ZO-1) after in vitro DENV-2 infection. The authors found that DENV infection caused a decreased expression and redistribution of both VE-cadherin and ZO-1 [57]. Moreover, many other molecules are released by activated ECs and have been strongly associated with severe plasma leakage in dengue infected patients. As such as a high level of syndecan-1, claudin-5, and chondroitin sulfate [58,59]. More recently, endoglin and syndecan-1 were quantified the mRNA expression and soluble protein levels in dengue cases during febrile, defervescence, and convalescence stages in DF, DFWS, and severe cases compared to non-dengue Other Febrile Illness (OFI) and healthy control. The authors observed a steady and significant increase in the levels of protein and mRNA of both the endoglin and syndecan-1 towards defervescence which is considered a critical phase in both severe and non-severe dengue cases. Importantly during the critical phase, the levels were significantly higher in Severe cases compared to DF, DFWS, and OFI controls [59]. It is possible that the days of illness also influenced the observation of a more significant change in the expression of adhesion molecules in HMVEC-d when adding the patients’ serum. Thus, it is important to evaluate patients with different clinical conditions, including the most severe ones, and to have a follow-up kinetics of the patients.

Interestingly, DF and DFWS/Sev sera decreased CD147/EMMPRIN expression on HMVEC-d and we also observed a decrease in MMP-9 levels in patients. CD147 (also known as basigin or EMMPRIN for extracellular matrix metalloproteinase inducer) is a membrane protein expressed in a variety of human tissues, including ECs. CD147 is involved in cell-matrix and cell-cell interactions, as well as ECM lysis and fibrosis [60]. ECM remodeling during its physiological turnover or during pathology schematically encompasses two opposing processes: ECM deposition (contraction) and ECM lysis. The latter is favored by proteinases, such as MMPs, which can be synthesized by various cell types, including resident cells (fibroblasts), transient cells (immune cells), or by nearby cells, such as invading cells in cancer [61]. To date, CD147 has been reported to induce the production of several extracellular MMPs, including MMP1, MMP-2, MMP-3, MMP-9, MMP-11, MT1-MMP, and MT2-MMP in different cell types and has been implicated in a variety of physiological and pathological activities [31]. More recently, Pan et al. revealed that DENV NS1 and MMP-9 induce cooperatively the vascular leakage by several mechanisms: NS1 recruits MMP-9 to degrade β-catenin, ZO-1, and ZO-2. Moreover, DENV activates the NF-κB signaling pathway to induce the expression of MMP-9 in the NS1 protein. Finally, the authors suggest that MMP-9 may act as a drug target for the prevention and treatment of DENV-associated diseases [62].

In our cohort, NS1 quantification did not differentiate the DF from DFWS/Sev or considering those above or below the “critical zone”. Thus, for us, the DENV NS1 and MMP-9 cooperatively mechanism of inducing vascular leakage is not appear more critical.

For us it was highly expected that sera from DFWS/Sev patients would decrease TEER compared to sera from DF patients. However, to our surprise, we have some sera from DF patients that also induced a decrease in TEER. Our hypothesis is that in mild dengue cases that presented a relative measure of TEER below the “critical zone”, this response could be involved with an endothelial alteration related to transendothelial migration of leukocytes, that is, activating an immunoprotected response. Other experimental approaches should be carried out to confirm this hypothesis and the inclusion of a greater number of critically ill patients would also help to clarify the observed changes.

Interestingly, high levels of IFN-γ and TNF-α were associated with a high relative TEER in our in vitro culture system, suggesting that the production of this set of cytokines could maintain the integrity of the endothelial cell monolayer, seen more clearly in DFWS/Sev patients. On the other hand, high levels of IFN-γ and IL-10 were related to a low relative TEER, suggesting that the production of this another set of cytokines could impair the integrity of the endothelial cell monolayer in DF patients. We do not know if there is a direct relationship between with our data comparing the Tian et al.’s data [60]. The authors identified a higher frequency of antigen-specific IL-10+IFN-γ+ double-positive (DP) CD4 T cells in during acute DHF than DF. The DP cells were characterized by non-cytotoxic/type 1 regulatory CD4 T cells and included IL-21, IL-22, CD109, and CCR1 [63]. However, the transcriptomic profile of DP cells was similar in DF and DHF, suggesting that DHF is not associated with the altered phenotypic or functional attributes of DP cells. Extrapolating for our data, quantitatively, the set IL-10/IFN-γ could contribute to vascular leakage in the severe forms of dengue.

The initial study by Chau et al. [64] demonstrated increased levels of several cytokines and chemokines, except CCL5/RANTES, in patients with DSS compared to DHF and patients with other febrile illnesses. At this point in the study, the authors suggested a relationship between low circulating levels of CCL5 and worse clinical signs of dengue. In cases of dengue death, our group demonstrated an increase in the frequency of CCL5+ cells in the patients’ liver, but lower circulating levels of CCL5 in the patients compared to healthy controls [65]. Another study also demonstrated an increase in the frequency of CCL5+ cells and viral antigens in liver, lung, and kidney tissues from dengue deaths [66]. Together, the studies indicate that high tissue expression of CCL5 and low circulating levels of CCL5 would be involved in cell migration to specific sites in dengue. In contrast, our studies point out that CCL5 was associated with maintenance of the integrity of the EC monolayer seen more clearly in DF patients. These data reinforce the idea that it is not just one or two cytokines, but there is a differential profile of sets of inflammatory mediators influencing endothelial alteration.

Recently, Wang et al. [67] performed in vitro assays comparing monolayer cultures of human umbilical vein endothelial cells (HUVECs) in 2-D tissue polystyrene plates and a 3-D culture model. Overall, the authors’ combined results demonstrate that the 3-D culture model allows for a better understanding of changes in cell morphology, oxidative stress, inflammatory cytokines, and endothelial function and is physiologically more relevant than 2-D. Certainly, 3-D culture will be a key point to advance understanding vascular changes in Dengue infection.

Our results are still very preliminary. Several elements, such as the NS1 DENV protein, proteins of the complement system, and other inflammatory mediators, influence the integrity of endothelial cells in dengue. However, obviously, it is impossible to assess “all these elements in an academic study”. We intend to evaluate representative samples of dengue patients for all four serotypes, from DENV-1 to 4. It is important to consider a broad discussion about the movement of the four DENV serotypes in the national territory and the year of entry of each serotype, which makes it difficult the availability a set of samples from patients infected by each of the serotypes at the same time. Also, to prove our hypothesis of the effect of IFN-γ, IL-10, and TNF-α in the regulation of endothelial integrity, we intend to demonstrate what happens after the treatment of endothelial cells with monoclonal antibodies against these mediators and verify if these treatments will reverse endothelial integrity. However, we believe that it will not be possible to show the effect of one or another isolated cytokine but of a subset of mediators that can influence endothelial integrity.

## 4. Materials and Methods

### 4.1. Study Population

The Viral Immunology laboratory team, during fieldwork in 2013, obtained 70 blood samples from dengue patients carried out at the Hospital Plantadores de Cana, at the Centro de Referência da Dengue located in Campos dos Goytacazes, Rio de Janeiro, and Hospital Dia Professora Esterina Corsini of the Universidade Federal do Mato Grosso do Sul, Mato Grosso do Sul. The samples of healthy controls were from laboratory staff and blood donors from the Hospital Universitário Clementino Fraga Filho of the Universidade Federal do Rio de Janeiro in the same time course.

We obtained the patients’ biochemical, hematological, and clinical signs/symptoms data from the medical records. Demographic, laboratory, and clinical characteristics of patients and controls are shown in Table 1.

### 4.2. Isolation and Cryopreservation of Serum Samples

About 10-mL venous blood samples were collected in serum tubes have a clot activator blasted into the tube wall that speeds up the coagulation process (BD Vacutainer^®^, Franklin Lakes, NJ, USA). The serum samples were stored at −70 °C until use.

### 4.3. Diagnosis

The following serological and/or virological tests were performed for dengue case confirmation: PanBio anti-dengue IgM and IgG (ELISA; PanBio, Brisbane, Australia and EL1500G Focus Diagnostics, Cypress, CA, USA); Platelia Dengue NS1 antigen (ELISA; Bio-Rad, Hercules, CA, USA) and molecular detection and serotype typing by real-time RT-PCR protocol [68,69]. Quantification of NS1 DENV was performed according to the protocol established by Heringer et al. [28]. Patients with primary infection were considered positive for any of the tests: anti-dengue IgM, DENV NS1 and/or a positive result by RT-PCR with negative anti-dengue IgG. Primary infection was characterized and considered by a positive anti-dengue IgG and IgM/IgG ratio > 2.0. Secondary infection, the IgM/IgG ratio was <2.0 [29].

### 4.4. Quantification of Serum Levels of TNF-α and CXCL8/IL-8 by Multiplex Methodologies in Liquid Microarray

The quantification of chemokines and cytokines was performed using serum from patients with a confirmed diagnosis of dengue. Dosages of IFN-γ, TNF-α, CXCL8/IL-8, and IL-10 were made by Multiplex assay in Liquid Microarray, by Luminex^®^ technology per manufacturer specifications (R&D Systems). This technique uses microspheres with defined spectral properties conjugated to capture antibodies specific for the target analytes. Specific microspheres for different targets are incubated simultaneously with a single serum sample, allowing the detection of multiple proteins in the same material. The average intensity of fluorescence is measured by the instrument and concentrations calculated in pg/mL. We used the Technology Platform of the Input Development Program to Health (PDTIS) (RPT03C-Luminex) by Fiocruz.

### 4.5. Quantification of Serum Levels of Soluble Mediators by ELISA

The quantification of chemokines and cytokines was performed using serum from patients with a confirmed diagnosis of dengue. ELISA assay was used to quantify CX3CL1/Fractalcina (catalog # DCX310, R&D Systems, Minneapolis, MN, USA), CXCL10/IP-10 (catalog # 900-T39, Peprotech, Rocky Hill, NJ, USA), CCL2/MCP-1 (catalog # 900-T31, Peprotech, Rocky Hill, NJ, USA), CCL5/RANTES (catalog # 900-T33, Peprotech, Rocky Hill, NJ, USA), and MMP-9 (catalog # KHC3061, Invitrogen, Carlsbad, CA, USA).

### 4.6. Cell Culture of Human Dermal Microvascular Endothelial Cells (HMVEC-d)

HMVEC-d cells (catalog #: CC-2543 Lonza, Basel, Switzerland) were propagated (passages 4–8) and maintained at 37 °C in humidified air with 5% CO_2_ in EGM^TM^ 2MV Microvascular Endothelial Cell Growth Medium-2 BulletKitTM (catalog #: CC-3202, Lonza, Basel, Switzerland) supplemented with growing factors, antibiotics, and 10% fetal bovine serum according to the manufacturer’s specifications.

### 4.7. Extracellular Staining by Flow Cytometry of Human Dermal Microvascular Endothelial Cells (HMVEC-d)

The expression of adhesion molecules and CD147 on HMVEC-d was determined by staining 100,000 cells with appropriate monoclonal antibodies (listed in Appendix A). Endothelial cells were initially suspended in PBS. After centrifugation, the cell pellet was resuspended in blocking solution (1% BSA, 0.1% NaN_3_, and 5% inactivated human plasma in PBS pH 7.2) for 30 min. Followed by incubation for 40 min 4 °C in the dark with monoclonal antibodies conjugated with fluorochrome. After incubation, the cells were washed twice with PBS and fixed in 1% paraformaldehyde. We kept fixed cells in the dark until acquisition. We acquired a minimum of 10,000 events in the region of cells considered viable on the BD Accuri™ (BD Bioscience, San Jose, CA, USA) cytometer. Analyzes were performed using Flow Jo v.7.6.1 software.

### 4.8. Transendothelial Electrical Resistance (TEER) Assay

HMVEC-d (2 × 10^5^ cells/well) in EGM^TM^ 2MV complete medium was seeded onto transwell inserts (3.0 µM, 6.5 mm insert; Corning Inc., Canton, NY, USA) and confluence was monitored by measuring transendothelial electrical resistance (TEER) across cell monolayers using Millicell-ERS system (Millipore, Billerica, MA, USA) voltmeter with chamber Endohm 6 (World Precision, Saratoga, FL, USA). The resistance values (TEER) were calculated as Ω/cm^2^ (Ohms) by subtracting the resistance value in each point time by the blank membrane resistance (without cells) and considering that resistance is inversely proportional to the area of the membrane (0.33 cm^2^). HMVEC-d grown in EGM^TM^ 2MV medium only on transwell inserts were used as time 0. On cell monolayers, 20% heat inactivated serum was added from patients in EGM^TM^ 2MV and cell permeability was monitored before (time 0) and after 5, 15, 30 min, 1, 2, 3, and 4 h. Data was normalized in relation to cells culture in EGM^TM^ 2MV only and relative TEER values at each kinetic point for each individual patient were calculated by understanding how many times the TEER value changed over time zero. In addition, we decided to set the threshold for the decrease in TEER (dashed a gray area 0.4 relative to TEER) constituted a “critical zone”, as all our healthy donors had relative TEER values above 0.4 considering at any time between 5 min to 4 h.

### 4.9. Statistical Analyzes

Non-parametric Mann–Whitney U tests and Kruskal–Wallis test followed by the Dunn post-test were performed were used to compare differences between study groups (Healthy donors vs. DF vs. DFWS/Sev). Nonparametric Friedman test followed by Dunn’s post-test was used to assess TEER measure checkpoints. For data were applied Wilcoxon matched pairs signed rank test. Correlation test was performed with non-parametric Spearman correlation correlogram. Statistical analyzes were performed using GraphPad Prism software, version 5.0 (GraphPad Software Inc., San Diego, CA, USA). *p* < 0.05 values were considered statistically significant.

## 5. Conclusions

Herein, acute dengue infection leads to an increase in the secretion of inflammatory mediators. Even so, only IL-10 distinguished mild to moderate cases of dengue, which would indicate its role in helping the clinical prognosis. DENV-4 infection and a little NS1 in the studied patients made it difficult to observe greater changes in the expression of adhesion molecules in ECs treated with patient serum. These mild conditions may also explain the decreased expression of CD147 in ECs and low levels of MMP-9 in the patients’ serum. It seems that different sets of inflammatory mediators would be related to the maintenance or loss of the integrity of EC monolayers. These results are still very preliminary. As a perspective, we will seek to understand whether there is a subset of cytokines and chemokines that, depending on this combination, can induce opposite effects regarding the effects on endothelial permeability and, consequently, on the clinical evolution of patients.

## Figures and Tables

**Figure 1 pathogens-11-00509-f001:**
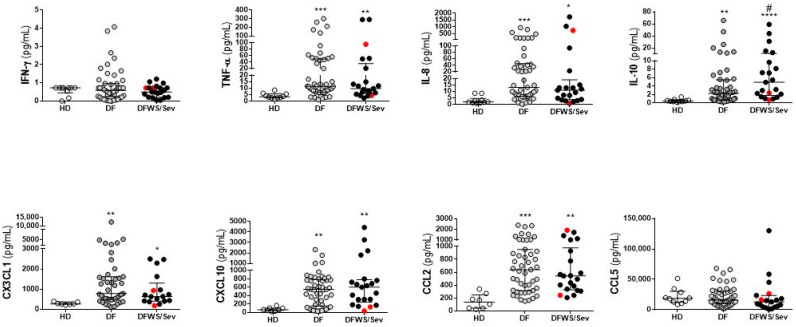
Cytokines and chemokines measurements in dengue patients. Serum samples from non-dengue healthy donors (n = 10), acute DF (n = 49), and acute DFWS/Sev (n = 21). Severe patients were identified by the red dots. The scatter dot plots show the median (middle line), and the interquartile range. The *p* values were calculated using the Kruskal–Wallis’s test followed by Dunn’s multiple comparisons test. Asterisks indicate significant differences between HD versus dengue patients (* *p* < 0.05, ** *p* < 0.01, *** *p* < 0.001 and **** *p* < 0.0001). The *p* values were calculated using the Mann–Whitney test and hashtag indicates significant differences between DF versus DFWS/Sev (# *p* < 0.05).

**Figure 2 pathogens-11-00509-f002:**
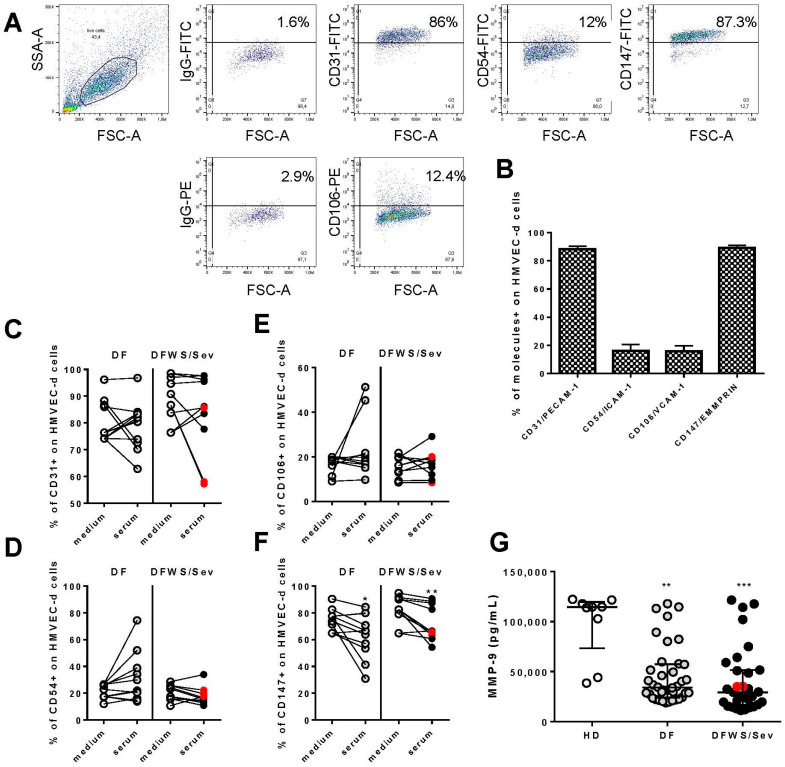
Expression of adhesion molecules CD31/PECAM-1, CD54/ICAM-1, CD106/VCAM-1, and CD147/EMMPRIN on HMVEC-d. (**A**) Figures representing the dot plot of morphological regions defined by size (FSC), *x*-axis and granularity (SSC), *y*-axis, of HMVEC-d cultured in supplemented culture medium. Marking dot plots of HMVEC-d with CD31, CD54, CD106, and CD147 and their isotype control are showed. Mean and standard deviation of the frequency of cells expressing each of the molecules of interest in HMVEC-d (**B**). From (**C**–**F**), cultures of HMVEC-d to which patient sera or culture medium were added and after 20 h, molecules expression were evaluated by Flow Cytometer. For data were applied Wilcoxon matched pairs signed rank test. For *, ** and *** indicate statistical significance with *p* < 0.05, *p* < 0.01 and *p* < 0.001 respectively. Severe patients were identified by the red dots. In (**G**), the measurements of MMP-9 in the serum of dengue patients.

**Figure 3 pathogens-11-00509-f003:**
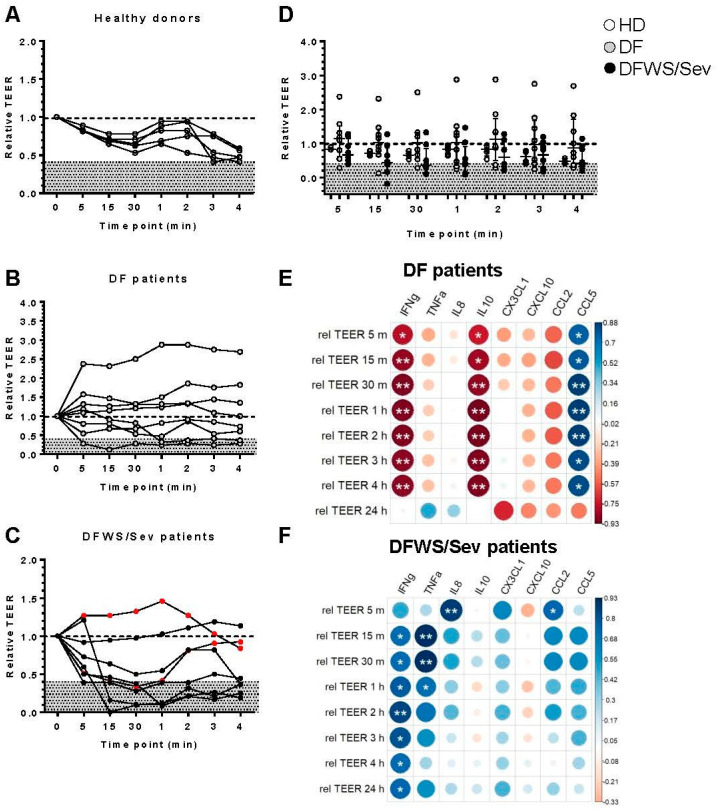
Transendothelial electrical resistance (TEER) measurement in confluent HMVEC-d monolayers of treated in the presence of 20% serum from dengue patients. Serum from healthy donors were used as controls (**A**). Eight DF (**B**) and 8 DFWS/Sev (6 DFWS and 2 Sev) (**C**) were randomly chosen. The transendothelial electrical resistance (TEER) was measured at 5, 15, and 30 min and 1, 2, 3, and 4 h. All TEER checkpoints were compared to time 0, defined as the time before the addition of serum. At time 0, we assumed the relative TEER values to be 1.0, and all TEER values were normalized. Statistical analyses were performed using Friedman’s test followed by Dunn’s multiple comparisons test between the TEER points. Comparing healthy donors, DF and DFWS/Sev patients in each timepoint of relative TEER values (**D**). The *p* values were calculated using the Kruskal–Wallis’s test followed by Dunn’s multiple comparisons test in each TEER point. Severe patients were identified by the red dots. Spearman correlation correlogram between relative TEER and cytokines/chemokines of cases of DF (**E**) and DFWS/Sev patients (**F**). The strength of the correlation between two variables is represented by the color of the circle, colors range from bright blue (strong positive correlation; rs = 1.0) to bright red (strong negative correlation; rs = −1.0); *p*-value is represented by asterisk (* *p* < 0.05, ** *p* < 0.01).

**Table 1 pathogens-11-00509-t001:** Demographic, clinical, and laboratorial data of study participants.

Characteristics	HD	DF	DFWS/Sev	*p* Value
Gender (Male:Female) ^a^	3:7	22:27	11:10	ns
Age, years ^b^	29 (25.3–30.3)	38 (23.5–54)	40 (24–55)	ns
days after the onset of symptoms ^b^	-	4 (2–6)	7 (4–9)	<0.004
Vascular changes ^a^	-	0	35	<0.0001
Bleeding ^a^	-	0	31.6	0.0003
Infecting serotype detection ^a^		48.8	21.1	ns
DENV-1 ^a^	-	4.8	0	ns
DENV-2 ^a^	-	0	25	ns
DENV-4 ^a^	-	95.2	75	ns
IgM anti-DENV ^a^	-	41.7	95.2	<0.0001
NS1 DENV Ag ^a^	-	64.6	50	ns
NS1 DENV Ag (ng/mL) ^b^	-	3.308 (2.165–5.111)	4.012 (1.599–5.074)	ns
Platelet count ^b,c^	276 (247.5–314.3)	164.5 (131–216.8) *	170 (111.5–201.8) *	<0.001
Hematocrit ^b^	40 (38.3–43.6)	39.4 (37.9–41)	39.7 (35.9–44.7)	ns
ALT (IU/L) ^b^	-	51 (36–76)	48 (30–198)	ns
AST (IU/L) ^b^	-	38.5 (28.8–63)	40 (30–128)	ns
Leucocyte count ^b,c^	6000 (5350–6080)	4050 (2870–5313)	3300 (2640–4900) *	<0.03
Lymphocyte count ^b,c^	1769 (1540–2073)	1172 (938.5–1654)	1097 (893.8–1261) *	<0.03
Monocyte count ^b,c^	402 (319.5–600)	440 (328–624)	327.5 (237–478.5)	ns

Study population n = 80; HD n = 10 (healthy donors), DF n = 49 (dengue fever without warning signs), and DFWS/Sev n = 21 (dengue fever with warning signs/severe dengue); ^a^ Gender, vascular leakage, hemorrhage, infecting serotype detection, IgM anti-DENV, and NS1 DENV Ag are given in positive/tested (%); Fisher’s exact test nonparametric test was applied. ^b^ Age, days after the onset of symptoms, NS1 DENv Ag, platelet number, hematocrit, transaminases, leukocyte count, lymphocyte count, and monocyte count are given in median (25–75%); ^c^ (×10^3^/mm^3^). For age, platelet number, hematocrit, leukocyte count, lymphocyte count, and monocyte count were applied Kruskal–Wallis’s test followed Dunn’s multiple comparisons nonparametric test. For days after the onset of symptoms and transaminases were applied Mann–Whitney nonparametric test. Bleeding included rash, petechiae, and gingival bleeding; Vascular changes included dyspnea, ascites, pleural effusion, and pericardial effusion; * Indicates statistical significance with *p* < 0.05. ns, not significant.

**Table 2 pathogens-11-00509-t002:** Demographic, clinical, and laboratorial data of Dengue patients.

Characteristics	TEER > 0.4	TEER ≤ 0.4	*p* Value
Gender (Male:Female) ^a^	4:4	2:6	ns
Age, years ^b^	33.5 (20.3–46.5)	42.5 (24.5–53.5)	ns
days after the onset of symptoms ^b^	4.5 (2.5–7.8)	5.5 (4.0–10.8)	ns
Vascular changes ^a^	25	16.7	ns
Bleeding ^a^	14.3	16.7	ns
Infecting serotype detection ^a^	37.5	62.5	ns
DENV-1 ^a^	0	0	ns
DENV-2 ^a^	0	0	ns
DENV-4 ^a^	100	100	ns
IgM anti-DENV ^a^	50	75	ns
NS1 DENV Ag ^a^	85.7	75	ns
NS1 DENV Ag (ng/mL) ^b^	4.461 (2.950–5.221)	4.317 (1.979–4.987)	ns
Platelet count ^b,c^	180 (154–246.5)	155 (120.5–194.5)	ns
Hematocrit ^b^	41 (40–45)	38.9 (33.8–40.1) *	<0.04
ALT (IU/L) ^b^	40 (18.5–138.8)	60 (48.8–1026)	ns
AST (IU/L) ^b^	31.5 (19.8–101)	67.5 (62.3–943.5)	ns
Leucocyte count ^b,c^	5070 (2923–7543)	3000 (1700–3400)	<0.06
Lymphocyte count ^b,c^	1199 (909.3–1829)	872.5 (637.5–1253)	ns
Monocyte count ^b,c^	651 (398.3–833)	300 (214.5–462)	<0.05
IFN-γ ^d^	0.41 (0.05–1.37)	0.62 (0.45–1.39)	ns
TNF-α ^d^	27.38 (4.46–183.4)	6.88 (4.14–9.30)	ns
CXCL8/IL-8 ^d^	22.45 (7.27–674.1)	5.42 (2.24–10.35)	<0.07
CXCL10/IL-10 ^d^	2.17 (1.12–4.75)	12.81 (2.10–36.82)	ns
CX3CL1/Fractalkine ^d^	1527 (936.4–2502)	761.1 (327.8–1707)	ns
CXCL10/IP10 ^d^	536.2 (56.9–872)	618.9 (368.8–730.8)	ns
CCL2/MCP-1 ^d^	942.7 (443.8–1414)	759.8 (355.3–1115)	ns
CCL5/RANTES ^d^	18,576 (7672–26,816)	5926 (3548–11,870)	<0.09

Study population n = 8; TEER > 0.4 n = 8 (dengue patients whose relative TEER value was above the “critical zone” of 0.4, TEER > 0.4.) and TEER ≤ 0.4 n = 8 (dengue patients whose relative TEER value was below or equal to the “critical zone” of 0.4); ^a^ Gender, vascular leakage, hemorrhage, infecting serotype detection, IgM anti-DENV, and NS1 DENV Ag are given in positive/tested (%); Fisher’s exact test nonparametric test was applied. ^b^ Age, days after the onset of symptoms, NS1 DENv Ag, platelet number, hematocrit, transaminases, leukocyte count, lymphocyte count, and monocyte count are given in median (25–75%); Mann–Whitney nonparametric test was applied. ^c^ (×10^3^/mm^3^). ^d^ pg/mL. Bleeding included rash, petechiae, and gingival bleeding; Vascular changes included dyspnea, ascites, pleural effusion, and pericardial effusion; * Indicates statistical significance with *p* < 0.05. ns, not significant.

## Data Availability

Not applicable.

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
