# Peer review of "Subsets of Cytokines and Chemokines from DENV-4-Infected Patients Could Regulate the Endothelial Integrity of Cultured Microvascular Endothelial Cells"

_pathogens, 2022, doi:10.3390/pathogens11050509_

Round 1

Reviewer 1 Report

Summary:

The authors studied the serum of cohort of Dengue Fever patients and looked at the differences between the dengue fever without warning signs (DF) group and the DFWS/Sev group including dengue fever with warning signs (DFWS) and severe dengue (Sev) fever. The study focused on presence of chemokines and cytokines and the correlation with the severity of dengue fever. Specifically, the authors try to argue that the presence of these cytokines/chemokines impact permeability of endothelial cells (EC). Right now, the study lacks multiple controls, and the data presentation deeply lacks in quality (see below many figures mistakes). The authors need to provide in depth revision of the figures. The presence of DENV NS1 protein should obviously be tested in the serum as it could alter all conclusions proposed by the authors. A strong rationale and extensive data points of control should be used for the “critical TEER zone” proposed by the authors as most data shown are not statistically significant.

Major comments:

  • Is 2D cell culture the best way to investigate endothelial integrity? In particular, 3D culture of endothelial cells is fairly easy to do and help further investigate tubule formation. In this context it would be very valuable to the authors arguments to perform some preliminary test in 3D culture and would significantly improve the significance of the study.

  • How would you explain the large difference of anti-DENV IgM between the DF and DFWS/Sev groups (41% vs 95% respectively)? Statistically it appears extremely significant (p<0.0001). However, the authors did not comment on it. A sentence addressing that should be added in the result section.

  • Figure 1: Is DENV NS1 protein present in the serum?

  • Figure 2B/E/F: There is a big mistake here. There are very large discrepancies between the levels of CD106- and CD147-positive EC in the bar graph 2B and the plots 2E-2F! In 2E plot, CD106+ is around 15-20% for all controls, whereas in 2B the baseline shows 85% CD106-positive. On the other hand, in 2F plot, CD147+ is around 65-90% for all controls whereas in 2B the baseline shows 15% CD147-positive. Are these two populations switched??? From the manuscript lines 184-186, it seems the mistake is limited to figure 2B. Please address this and amend the manuscript as needed!

  • Figure 2C/D/E/F: Why is there 3 red dots in plots C and D? There are only 2 severe patients in the cohort so there should only be 2 red dots.

In addition, the number of sample do not match the text: Lines 176-177, the authors stipulated that EC were “cultured in the presence of randomly chosen sera of 11 DF and 10 DFWS/Sev (8 DFWS and 2 Sev) patients”. Therefore, there should be 10 dots/samples for DF samples and 11 dots/samples for DFWS/Sev samples. It definitely does not appear this way. Please provide the raw data and analysis used for all the dots presents in figures 2C/D/E/F.

  • Lines 189-190: Is there a difference between DF and DFWS/Sev for CD147 expression? The author should clearly stipulate if there is or not.

  • Line 191-195: Authors should clearly present in a figure the correlation they claim significant between CD147 and CX3CL1/TNF-a.

  • Figure 3A: Why are so many data points missing? Healthy control is missing all 24h treatment data points. Samples data points randomly start missing after 2 or 4hrs for all conditions. This need to be addressed comprehensively. Why are these data points missing? Are they removed? Or cell being unhealthy? If so, is the data generated trustable.

  • Lines 227-234: This analysis seems farfetched and very biais. Can you cite any previous study that do such inference? Can the authors provide the data used for the statistics and the exact test used? Comparison would have to stop at 4h treatment as no baseline (healthy donor) is shown for 24h treatment time point. Is there healthy donor data points at 24h and are they in the “TEER critical zone”? As the trend is clearly decreasing…

  • Lines 235-236, I disagree with the conclusion that there are changes in the permeability of EC.

  • Figure 3E and F should use the same color scheme scale/unit for comparison. Right now, the data and labels are not clear enough to see if that’s the case.

  • Figure 3: Is NS1 present in the serum? It is critical to know if NS1 is present or not as the protein is highly inflammatory and impact EC permeability as the authors themselves stipulate in the introduction (lines 65-75). This could all potentially be explained by NS1.

  • To be convincing the authors should perform validation experiments either. For instance, by adding the chemokines/cytokines proposed to be impactful directly to the serum of healthy donors and culture EC in it. Right now, the conclusions are correlational at best and could be consequence to other protein in the serum such as viral NS1.

Minor comments:

  • Lines 57-60: The sentence needs to be rephrased. In particular the section/line59 “with a consequent reduction in blood pressure and pulses well as impaired perfusion organic”.

  • Figure 1: Second Sev patient RED dot is not visible in the TNF-a and IL-8 and should be made visible (bring to foreground), or did the author remove the data point?

  • Figure 2A: The gating strategy should be put as a supplementary figure, and Figure 2A should instead be the histogram graphs. It would look much neater. However gating strategy and histogram for CD147 are missing and should be added as well.

Reviewer 2 Report

In this study, Cipitelli et al. studied the effect of sera from patients with mild and severe dengue on the integrity of endothelial cells and the expression of adhesion molecules (CD31, CD54, CD106, and CD147). The authors also evaluated the serum levels of some soluble mediators in these patients and associated them with changes observed in the permeability of endothelial cells.

There were differences in the levels of cytokines/chemokines found in the circulation of patients with mild and severe dengue. In addition, there were differences in the permeability of endothelial cells induced by serum from subjects that experienced severe dengue compare to mild dengue.

My concern is that serum from some subjects contains NS1 and dengue-specific antibodies, which can contribute to the effects seen in endothelial cells. In this study, there were no block or addition of cytokines to the endothelial cell monolayers to confirm their effects. In addition, there was no removal of NS1 or dengue-specific antibodies from the serum samples. If possible, I would recommend the authors include the limitations of the study in the discussion section.  

  • I like the idea of identifying the levels of serum soluble mediators in severe patients with red dots in figure 1, however, some of them are not distinguished.
  • Were the serum samples heat inactivated?
  • In my opinion, the colors of the symbols in Figures 3A, B, and D need to be changed.
  • Figure 3A-C may need to include the average.
  • In my opinion, the discussion is good but should be shorted.

Round 2

Reviewer 1 Report

Authors made significant effort to address my comments by explaining contentious points, correcting the problems with the figures and providing additional supplemental material (tables and figures) to complete the manuscript.

I only have one minor suggestion related to presentation:

- Right now, Table 1 is cut in the middle over 2 pages and the authors might want to reposition it to avoid that.

Reviewer 2 Report

Thank you to the authors for addressing most of the questions and concerns. This new version extends the limitations of the study and in my opinion, the manuscript is suitable for publication.